# First clinical expression of equine insect bite hypersensitivity is associated with co-sensitization to multiple *Culicoides* allergens

Jasmin Birras[1], Samuel J. White[2,3], Sigridur Jonsdottir[1,4], Ella N. Novotny[1], Anja Ziegler[1], A. Douglas Wilson[5], Rebecka Frey[6], Sigurbjörg Torsteinsdottir[4], Marcos Alcocer[3], Eliane Marti[1] *

1 Department of Clinical Research and Veterinary Public Health, Vetsuisse Faculty, University of Bern, Bern, Switzerland, 2 School of Animal, Rural and Environmental Sciences, Nottingham Trent University, Brackenhurst Campus, Southwell, United Kingdom, 3 School of Biosciences, University of Nottingham, Loughborough, United Kingdom, 4 Institute for Experimental Pathology, Biomedical Center, University of Iceland, Keldur, Reykjavik, Iceland, 5 Division of Veterinary Pathology, Infection and Immunity, University of Bristol, Langford, United Kingdom, 6 AniCura Norsholms Djursjukhus, Norsholm, Sweden

* eliane.marti@vetsuisse.unibe.ch

**Data Availability Statement:** All relevant data are within the manuscript and its Supporting Information files.

## Abstract

### Background

Insect bite hypersensitivity (IBH) is an IgE-mediated allergic dermatitis in horses incited by salivary allergens from *Culicoides spp*. IBH does not occur in Iceland, as the causative agents are absent, however a high prevalence is seen in horses exported to *Culicoides*-rich environments.

### Aims

To study the natural course of sensitization to *Culicoides* allergens and identify the primary sensitizing allergen(s) in horses exported from Iceland utilizing a comprehensive panel of *Culicoides* recombinant (r-) allergens.

### Method

IgE microarray profiling to 27 *Culicoides* r-allergens was conducted on 110 serological samples from horses imported to Switzerland from Iceland that subsequently developed IBH or remained healthy. Furthermore, a longitudinal study of 31 IBH horses determined IgE profiles the summer preceding first clinical signs of IBH ($T_{IBH}$-1), the summer of first clinical signs ($T_{IBH}$) and the following summer ($T_{IBH}$+1). In a group of Icelandic horses residing in Sweden, effects of origin (born in Iceland or Sweden) and duration of IBH (<4 years, 4–7 years, >7 years) on *Culicoides*-specific IgE was evaluated. Sero-positivity rates and IgE levels were compared.

### Results

At $T_{IBH}$, horses were sensitized to a median of 11 r-allergens (range = 0–21), of which nine were major allergens. This was significantly higher than $T_{IBH}$-1 (3, 0–16), as well as the

**Funding:** This study was supported by the Swiss National Science Foundation grant no 310030-160196/1, by the Morris Animal Foundation grant no D16EQ-039 and by the Stiftung Forschung für das Pferd and by the Icelandic Research Fund grant no 141071-05. The funders had no role in study design, data collection and analysis, decision to publish, or preparation of the manuscript.

**Competing interests:** The authors have declared that no competing interests exist.

**Abbreviations:** AIT, allergen-specific immunotherapy; Alt a 1, recombinant mould allergen *Alternaria alternate 1*; CN-TE, *Culicoides nubeculosus* thorax extract; CO-WBE, whole body extract from *Culicoides Obsoletus* group midges; cP, corrected p-value; Cul n, *Culicoides nubeculosus*; Cul o, *Culicoides obsoletus*; Der f, house dust mite extract (*Dermatophagoides farinae*); FAU, fluorescence arbitrary unit; H, healthy; IBH, insect bite hypersensitivity; IS, born in Iceland and exported to Sweden; N-IS, born in Sweden and living in Sweden; ns, not significant; ROC, Receiver Operator Characteristic; SV-WBE, black fly extract (*Simulium vittatum*); $T_{IBH}$, summer of first clinical signs of IBH; $T_{IBH}+1$, year following summer of first clinical signs of IBH, i.e. 2nd summer with clinical signs of IBH; $T_{IBH}-1$, year preceding summer of first clinical signs of IBH; unexposed, horses living in Iceland, not exposed to *Culicoides*.

healthy (1, 0–14) group. There was no significant increase between $T_{IBH}$ and $T_{IBH}+1$(12, 0–23). IBH-affected horses exported from Iceland had a significantly higher degree of sensitization than those born in Europe, while duration of IBH did not significantly affect degree of sensitization.

## Conclusion

Significant sensitization is only detected in serum the year of first clinical signs of IBH. Horses become sensitized simultaneously to multiple *Culicoides* r-allergens, indicating that IgE-reactivity is due to co-sensitization rather than cross-reactivity between *Culicoides* allergens. Nine major first sensitizing r-allergens have been identified, which could be used for preventive allergen immunotherapy.

## Introduction

Insect bite hypersensitivity (IBH), also known as *Culicoides* hypersensitivity, is the most common allergic skin disease in horses. IBH is a seasonal allergic dermatitis caused by hypersensitivity reactions to bites of blood feeding insects of the genus *Culicoides*. Clinical signs are mainly seen in the mane and tail area and derive from severe pruritus which leads to hair loss and excoriations and the development of chronic skin lesions, and sometimes to secondary infections [1, 2].

This disease occurs in all breeds, with a prevalence of 3–10% across Europe. Horses living in Iceland do not suffer from IBH as horse-biting *Culicoides* species are absent. However, >50% of Icelandic horses exported to continental Europe as adults develop IBH within the first 2 years post *Culicoides* exposure, while Icelandic horses born in Europe do not have a higher prevalence of IBH than other breeds [2, 3]. Interestingly, horses exported from Iceland and exposed to *Culicoides* before seven months of age have the same low risk of developing IBH as locally bred horses, suggesting that early exposure to *Culicoides* allergens is essential for the development of immune tolerance [1, 4]. Allergen-specific immunotherapy (AIT) is the only causative treatment for type I hypersensitivities, leading to a shift from a Th2 immune response to a regulatory immune response, in which IgG antibodies are produced that block allergen specific IgE antibodies binding to allergens [5]. Currently, the efficacy of AIT treatment of IBH is questionable, as placebo controlled studies could not demonstrate an effect of AIT compared to the placebo group [6, 7]. This lack of efficacy is most likely due to the fact that crude *Culicoides* whole body extracts were used instead of pure allergens [2]. With the aim to improve AIT and diagnostic serology for IBH, molecular approaches have been applied for the identification of *Culicoides* salivary allergens and their production as recombinant (r-) proteins [2]. Within the last decade, 30 *Culicoides* salivary allergens have been produced as recombinant proteins derived from three *Culicoides* species: *C. obsoletus* [8–10] *C. nubeculosus* [11, 12] and *C. sonorensis* [13]. Microarray profiling of horses from various breeds living in central and northern Europe utilising 27 *C. nubeculosus* and *C. obsoletus* r-allergens identified nine major *Culicoides* r-allergens, seven of which bound IgE in sera of >70% of IBH-affected horses. The combination of these seven allergens could correctly diagnose >90% of IBH-affected horses with a specificity of ≥95% [10]. All nine major allergens were derived from *C. obsoletus*, confirming that allergens derived from *Culicoides* species present in the horse's environment are more immune-reactive than laboratory-bred species [9, 14].

While the use of AIT is well-established for the treatment of human allergies, its use as preventive immunotherapy in high-risk individuals has been proposed prior to sensitization [15–17]. Whether preventive AIT against IBH is feasible in horses remains to be established, and might become an interesting option to decrease the high incidence of IBH in horses exported from Iceland to continental Europe. However, identification of the primary sensitizing allergens is a prerequisite before such experiments can be performed.

Hence, the objectives of this study were to characterize the natural course of sensitization to *Culicoides* allergens, and identify the primary sensitizing *Culicoides* r-allergen(s) for IBH in horses exported from Iceland to Switzerland. Allergen-specific IgE levels to a large panel of *Culicoides* allergens were determined by protein microarray using sera from a longitudinal study. Additionally, effects of the horse's origin and duration of IBH (years with IBH) on the pattern of sensitization and levels of allergen-specific IgE were investigated.

## Material and methods

### Horses and blood samples

A total of 224 adult horses of the Icelandic breed were included in the study (Table 1). One hundred and ten horses of which had been exported from Iceland, now residing in Switzerland [18]. Fifty-one horses had remained free from IBH (group H), while 59 developed IBH after export (group IBH). This group was monitored over a period of at least three summers, starting at the time of importation into Switzerland [18]. Longitudinal samples were collected from 31 of the 59 horses i.e. serum IgE levels were analyzed one year before first clinical signs of IBH ($T_{IBH}$-1), the year when IBH occurred for the first time ($T_{IBH}$), as well as the following year ($T_{IBH}$+1). The serum samples from the 51 horses of the H end point group were selected to match the years of sampling in the IBH horses, i.e. the serum samples of the H matched the year of sampling of the IBH-horses at $T_{IBH}$. Forty one of these 51 control horses had been imported the same year as the IBH horses. No longitudinal study was performed on the H end-point group, as a previous study demonstrated there is no increase in allergen-specific IgE in sera from horses with a healthy end-point [19]. Additionally, sera from 22 horses living in Iceland were included in the study. These horses had been used in a previous study [20].

Data and sera from 92 horses living in Sweden (55 with IBH and 37 healthy controls) was also utilized [20, 21]. In 27 of the 55 Swedish IBH horses the duration of disease at time of blood sampling was known, and as such were grouped according to disease duration. Group 1 had clinical signs of IBH for < 4 years, group 2 for 4–7 year, and group 3 for > 7 years.

Horses defined as IBH affected showed the typical recurrent seasonal signs [1], while horses defined as healthy did not show clinical signs of IBH or other skin diseases.

Sera from horses living in continental Europe (exposed to *Culicoides*), affected with insect bite hypersensitivity (IBH) or healthy (H), as well as from healthy horses living in Iceland, thus not exposed to *Culicoides* bites (unexposed). All horses belong to the Icelandic breed. For 31

**Table 1. Horses included in the study.**

| Living in | Born in Iceland (IS) | | | Born in continental Europe (N-IS) | | Study |
|---|---|---|---|---|---|---|
| | unexposed | IBH | H | IBH | H | |
| Switzerland (N = 110) | - | 59 | 51 | - | - | Torsteinsdottir et al. 2018 [18] |
| Sweden (N = 92) | - | 44 | 13 | 11 | 24 | Frey et al. 2008 [20]; Heimann et al. 2011 [21] |
| Iceland (N = 22) | 22 | - | - | - | - | Frey et al. 2008 [20] |
| Total (N = 224) | 22 | 103 | 64 | 11 | 24 | |

IBH horses living in Switzerland, data from 3 consecutive time points was available: the year of first clinical signs of IBH ($T_{IBH}$), the preceding ($T_{IBH}-1$) and the following year ($T_{IBH}+1$).

The sera used in the study had been collected between May and November, i.e. during the IBH season. Blood was collected from the jugular vein using Serum Clot Activator-containing vacutainers (Vacuette®; Greiner, St.Gallen, Switzerland). Serum was separated and stored at -80˚C until analysis. The study was approved by the Animal Experimental Committee of the Canton of Berne, Switzerland (No. BE 121/05 and BE 2/17). Verbal owner consent was obtained for all horses included in the study.

## Serum IgE profiling by protein microarray

Determination of IgE in serum was performed by protein microarray, as previously described [10, 22, 23], using the same protein microarray as in Novotny et al. [10] which included a total of 27 *Culicoides* r-allergen, *Culicoides* and *Simulium vittatum* extracts, as well as proteins irrelevant for IBH as controls (Table 2).

Table 2. Allergens and cut-off values used in the study.

| Name | Expression System | Protein family | GenBank | Cut-off used (FAU) |
|------|------|------|------|------|
| Cul o 1P | Coli | Kunitz Protease Inhibitor | JX512273 | 7300 |
| Cul o 2 | Baculo | Hyaluronidase | KC339672 | 150 |
| Cul o 2P | Coli | D7-related/OBP | JX512274 | 300 |
| Cul o 3 | Coli | PR1 like (antigen-5 like) | KC339673 | 1070 |
| Cul o 3P | Coli | D7-related/OBP | JX512275 | 870 |
| Cul o 5 | Coli | Unknown | KC339675 | 6000 |
| Cul o 6 | Pichia | D7-related / OBP | KC339676 | 238 |
| Cul o 7 | Baculo | Unknown | KC339677 | 900 |
| Cul o 8 | Coli | Kunitz protease inhibitor | MN123710 | 1650 |
| Cul o 9 | Coli | WSC superfamily, carbohydrate binding domain | MN123712 | 1800 |
| Cul o 10 | Coli | DUF4803 superfamily | MN123711 | 700 |
| Cul o 11 | Coli | Apolipophorin III like | MN123713 | 11000 |
| Cul o 12 | Coli | Leucin rich repeat | MN123714 | 488 |
| Cul o 13 | Coli | D7-related/OBP | MN123715 | 5600 |
| Cul o 14 | Coli | Serine protease/Trypsin | MN123716 | 160 |
| Cul o 15 | Coli | Apyrase | MN123717 | 1100 |
| Cul n 1 | Baculo | PR1 like (antigen-5 like) | EU978899 | 2500 |
| Cul n 2 | Baculo | Hyaluronidase | HM145950 | 230 |
| Cul n 3 | Baculo | †DUF4803 superfamily | HM145951.1 | 7500 |
| Cul n 4 | Barley | Unknown | HM145952 | 820 |
| Cul n 5 | Coli | DUF4803 superfamily | HM145953 | 108 |
| Cul n 6 | Coli | unknown | HM145954 | 800 |
| Cul n 7 | Coli | Unknown | HM145955 | 2200 |
| Cul n 8 | Baculo | Maltase (alpha amylase) | HM145956 | 3950 |
| Cul n 9 | Coli | D7-related/OBP | HM145957 | 2400 |
| Cul n 10 | Coli | DUF4803 superfamily | HM145958 | 3500 |
| Cul n 11 | Coli | Serine Protease/Trypsin | HM145959 | 1500 |
| CN-TE | extract | | [19] | 1150 |
| SV-WBE | extract | | [18] | 1800 |
| CO-WBE | extract | | | 1550 |
| Alt a 1 | Coli | | Biomay | 300 |
| Der f | extract | | StallergeneGreer | 3700 |

These proteins had been normalized to 0.5 mg/ml protein and printed onto Grace Bio-Labs Oncyte® Nova™ nitrocellulose film slides using a Marathon microarrayer (Arrayjet, Roslin, Scotland) [22]. Slides were first blocked with 3% BSA in PBS. Sera, diluted 1:2, were applied and the slides hybridized O/N at 4˚C. After washing, anti-horse IgE mAb 3H10 [24] was added and incubated at 37˚C for 2h, followed by an incubation with DyLight 649 conjugated anti-mouse IgG1 for 1h (Rockland, #610-443-040). The slides were then dried via centrifugation and scanned using a GenePix4000B (Molecular Devices, Inc., Sunnyvale, CA, USA). For each protein blank values (obtained by adding all reagents except serum) and background fluorescence were subtracted from the values obtained with the sera before further analyses of the data. Data were presented as fluorescence arbitrary unit (FAU).

*Culicoides nubeculosus* (Cul n) and *Culicoides obsoletus* (Cul o) recombinant allergens and Cul n thorax extract (CN-TE), Cul o group whole body extract (CO-WBE) and *Simulium vittatum* whole body extract (SV-WBE) used in the study. Cut-off values used (in fluorescence arbitrary units, FAU) are those defined previously [10]. *Alternaria alternate* 1 (Alt a 1) and *Dermatophagoides farinae* (Der f) were used as control proteins not relevant for IBH.

## Statistical analyses

For statistical analyses, NCSS software (NCSS 12 Statistical Software (2018) NCSS, LLC. Kaysville, Utah, USA, ncss.com/software/ncss) was used. Since the data was not normally distributed, descriptive statistics using median and ranges were used. The non-parametric Kruskal-Wallis Multiple-Comparison Z-Value test (Dunn's test) with Bonferroni correction for multiple comparisons was used to analyze differences in allergen-specific IgE concentrations or numbers of positive IgE values between IBH-affected, healthy control and unexposed horse groups as well as between horses born in Iceland (IS) and born in continental Europe (N-IS), and horses grouped according to the duration of IBH.

For each allergen, specific IgE values were transformed to positive and negative (above and below cut-off level) results. As IgE measurement of these samples was carried out within few weeks of those from Novotny et al. [10], the same cut-off values were used (Table 2). Values giving at least a specificity of 94% at the highest accuracy possible had been selected as cut-offs [10].

Median values between time points in the longitudinal study were compared using the non-parametric paired Wilcoxon Signed-Rank Test. Bonferroni correction for multiple comparisons (cP) was performed manually (P-value x number of comparisons = cP).

The 2-sided Fisher's exact test was used to compare the proportion of IBH-affected, H horses and unexposed horses with positive allergen-specific IgE results, or to compare the proportion of positive results in horses born in Iceland and exported to Sweden or born and living in Sweden. When multiple comparisons were performed, cP was used as mentioned above. P ≤ 0.05 was regarded as significant throughout the paper.

## Results

### Allergen-specific IgE in sera from horses born in Iceland and imported to Switzerland

First, IgE levels to *Culicoides* allergens in sera collected the first summer of clinical onset of IBH were compared between the IBH (n = 59) and the H group (n = 51), as well as unexposed horses that were living in Iceland (n = 22).

Depending on the *Culicoides* r-allergen, the percentage of IBH horses with positive IgE values ranged from 5 to 78%. In the H group, 2 to 14% of the horses had positive IgE values, and

in the unexposed horses between 0 to 18% (Fig 1). There were no significant differences between the unexposed and H groups for any of the tested r-allergens. Surprisingly, there were some IgE values above cut off in the unexposed horses, however IgE concentrations were

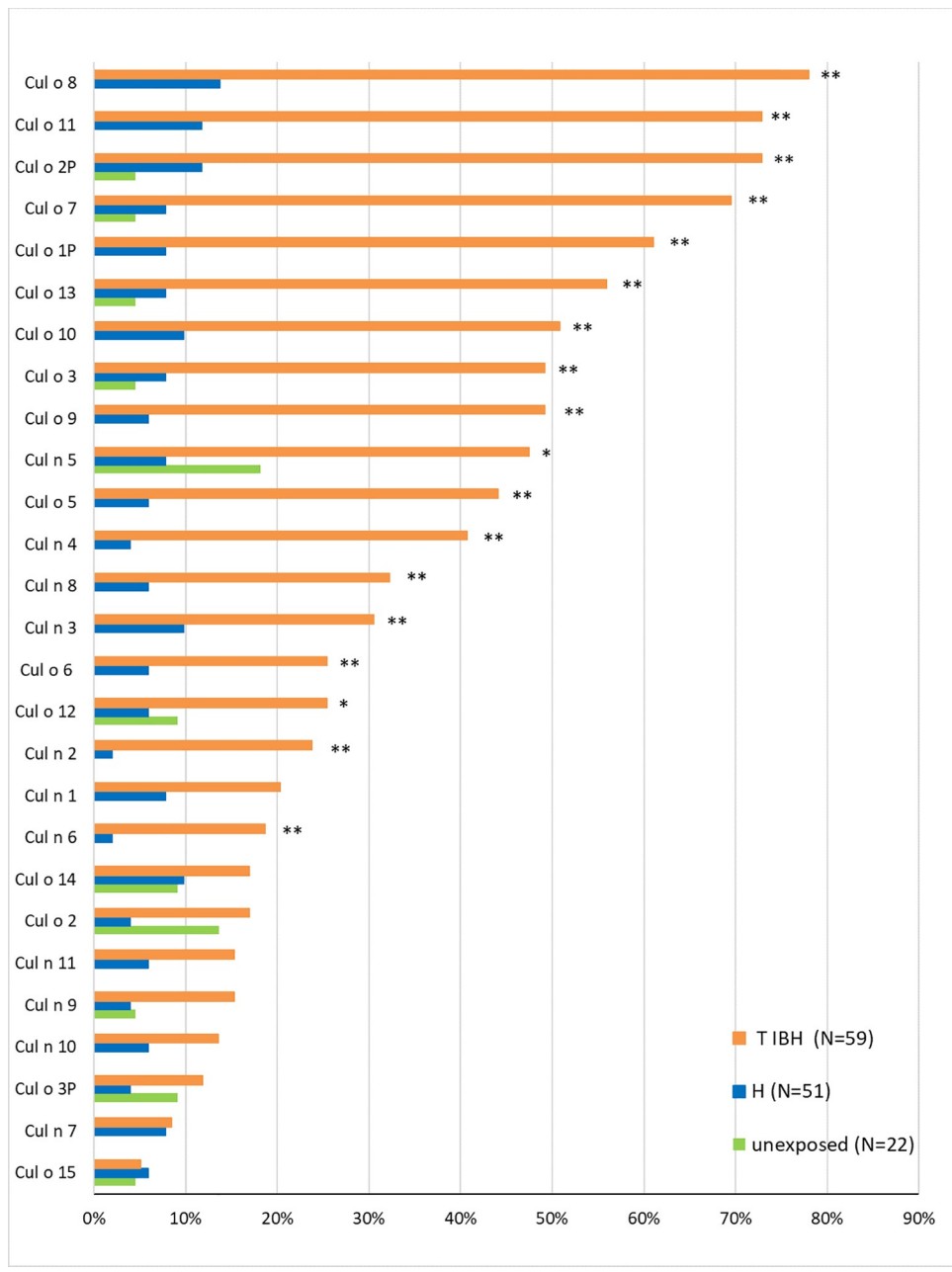

**Fig 1. IgE sero-positivity to 27 *Culicoides* recombinant (r-) allergens.** Percentage of horses with IgE levels above the cut-off values (as indicated in Table 2) in horses exported from Iceland to Switzerland that developed insect bite hypersensitivity (IBH; n = 59) or remained healthy (H; n = 51), and in horses living in Iceland (unexposed; n = 22). Serum samples were taken the summer of clinical onset of IBH ($T_{IBH}$) and at the corresponding time in the H group. The allergens are listed in decreasing order from those binding serum IgE in the highest number of horses at time of clinical onset of IBH ($T_{IBH}$). P values were calculated with the Fisher's exact test and Bonferroni correction done for multiple comparisons. ** IBH significantly different from H and unexposed (cP <0.05). * IBH significantly different from H (cP<0.05). No significant differences between H and unexposed for any r-allergen.

usually low (S1 Table). Eighteen of the 27 tested *Culicoides* r-allergens bound serum IgE in a significantly higher percentage of IBH-affected horses compared to the H horses (Fig 1). Seven of these allergens (Cul o 8, Cul o 11, Cul o 2P, Cul o 7, Cul o 1P, Cul o 13 and Cul o 10) bound serum IgE in >50% of the IBH-affected horses, and two in almost 50% of them (Cul o 3 and Cul o 9, each 49.2%). Fig 1 shows that the allergens Cul o 8, Cul o 11 and Cul o 2P bound serum IgE in > 70% of the IBH affected horses, Cul o 7 and Cul o 1P in > 60% and Cul o 13 and Cul o 10 in >50%.

Median IgE levels to the r-*Culicoides* allergens in the three groups of horses are shown in S1 Table. Interestingly, while there were no significant differences between H and IBH-horses for IgE concentrations specific for the irrelevant proteins (Der f and Alt a 1), the horses living in Iceland (unexposed) had significantly higher IgE to these allergens than those living in Switzerland.

## Longitudinal study of allergen-specific IgE in sera from horses that developed IBH

In a subgroup of 31 horses that developed IBH following import, IgE reactivity patterns to *Culicoides* r-allergens were determined at three different time points: the year preceding first clinical signs of IBH ($T_{IBH}$-1), the year when IBH occurred for the first time ($T_{IBH}$) and the following year ($T_{IBH}$-1). The number of positive IgE reactions to *Culicoides* r-allergens increased significantly between $T_{IBH}$-1 and $T_{IBH}$, from a median number of three (range 0–16) to eleven positive reactions (range 0–21), while there was no significant increase between $T_{IBH}$ and $T_{IBH}$+1. At this last time point, IBH horses had positive IgE values to a median number of 12 (range 0–23) *Culicoides* r-allergens (Fig 2).

At $T_{IBH}$-1, horses that developed IBH did not show significantly higher sero-positivity rates to individual r-allergens than the H group, except for Cul n 2 and Cul n 11 (Fig 3). However, for both of these allergens the sero-positivity rate remained low over time (<33%).

Analysis of the reactivity to the individual *Culicoides* r-allergens demonstrates that the number of positive IgE reactions increased significantly between $T_{IBH}$-1 and $T_{IBH}$ for ten Cul o r-allergens, as well as for Cul n 5 (Fig 3). At $T_{IBH}$, the allergens Cul o 11, 8, 7, 2P, 10, 1P and 13 bound IgE in >50% of the IBH sera. Even though comparison of the percentage of IgE positive reaction between $T_{IBH}$ and $T_{IBH}$+1 did not reach statistical significance for any *Culicoides* r-allergen, for many r-allergens a further increase in the number IgE positive horses was observed. At $T_{IBH}$+1 >80% of the horses had positive IgE values with Cul o 8 and Cul 2P, >70% with Cul o 11, Cul o 7, Cul o 10 and Cul o 1P and >60% with Cul o 13.

IgE levels were compared using a paired T-test to test whether the amount of free serum IgE to a given allergen increased in the single horses over time (Table 3). As expected from the previous analysis (Fig 3), for most r-allergens the main increase in IgE levels was observed between $T_{IBH}$-1 and $T_{IBH}$ (Table 3). Between $T_{IBH}$ and $T_{IBH}$+1 a significant increase in IgE levels was observed only for Cul o 1P, Cul o 8, Cul o 9 and Cul n 10.

## Comparison of allergen-specific IgE in sera of horses from the Icelandic breed with different origins

The influence of the origin of the horse (i.e. born in Iceland and exported to Sweden or born in Sweden) on *Culicoides* r-allergen specific IgE was evaluated in a group of horses located in Sweden. In horses born in Iceland and exported to Sweden, IBH-affected horses had significantly higher median IgE levels to 20 different *Culicoides* r-allergens compared to H horses. Within horses born in Sweden, IBH-affected horses had significantly higher IgE levels to eight of these *Culicoides* r-allergens (Table 4). Moreover, IgE levels did not differ significantly

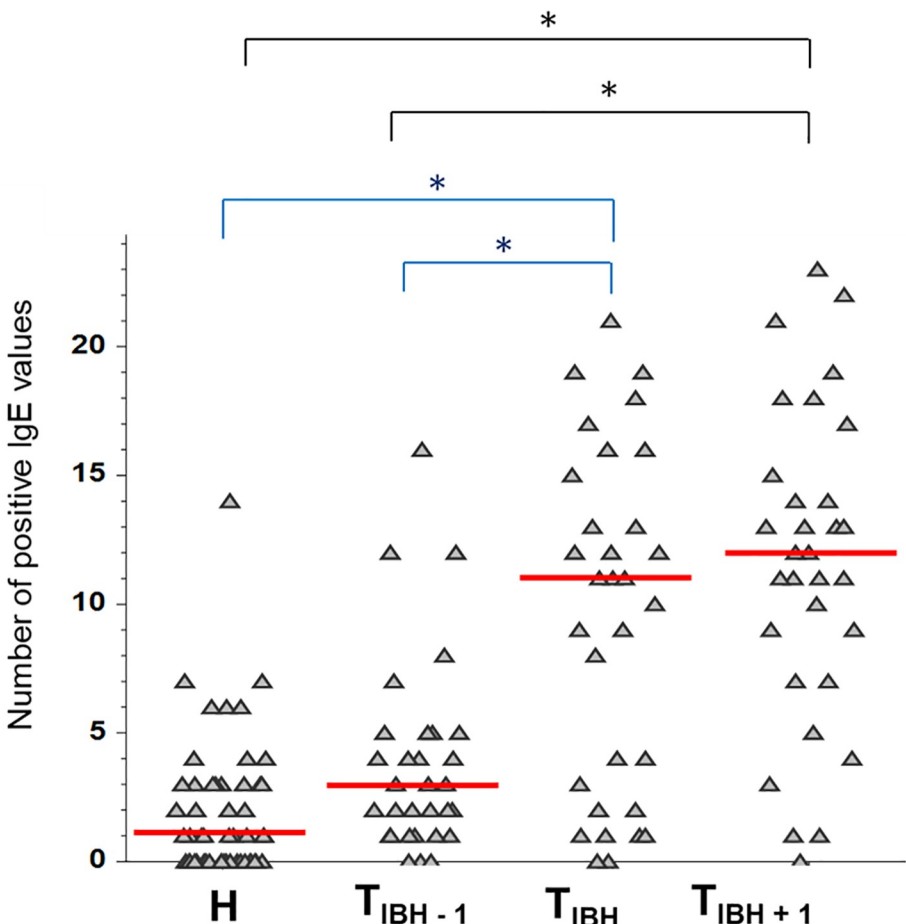

**Fig 2. Cumulative number of positive serum IgE values per horse for the 27 r-allergens at time of clinical onset of IBH (T$_{IBH}$), one year before (T$_{IBH}$-1) and one year after (T$_{IBH}$+1) and in healthy control horses (H).** Each symbol represents a separate horse and red lines represent the medians. The Kruskal-Wallis Multiple-Comparison Z-Value test (Dunn's test) with Bonferroni correction for multiple comparisons was used. * indicates significant differences between the time points (P ≤ 0.01).

between H horses born in Sweden or H horses born in Iceland and exported to Sweden. Horses exported from Iceland that later developed IBH had significantly higher median IgE levels to Cul o 6, Cul o13 and Cul n 5 than IBH horses born in Sweden. Particularly Cul o 13 seems to be of high importance in IBH-affected Iceland-born horses, while not in continental-born IBH horses.

## Effect of duration of IBH on allergen-specific IgE levels

To investigate whether horses affected with IBH for several years have higher IgE levels and react to a higher number of r-allergens, IBH-affected horses living in Sweden were grouped according to disease duration. Horses suffering from IBH for <4 years had IgE positive to a median number of 15 r-allergens (range 4–22). This value remained the same in horses affected with IBH for a duration of 4 to 7 years (median 15, range 6–22) and increased to 18 (range 8–21) r-allergens for horses suffering from IBH for more than 7 years, but this difference was not significant (Fig 4).

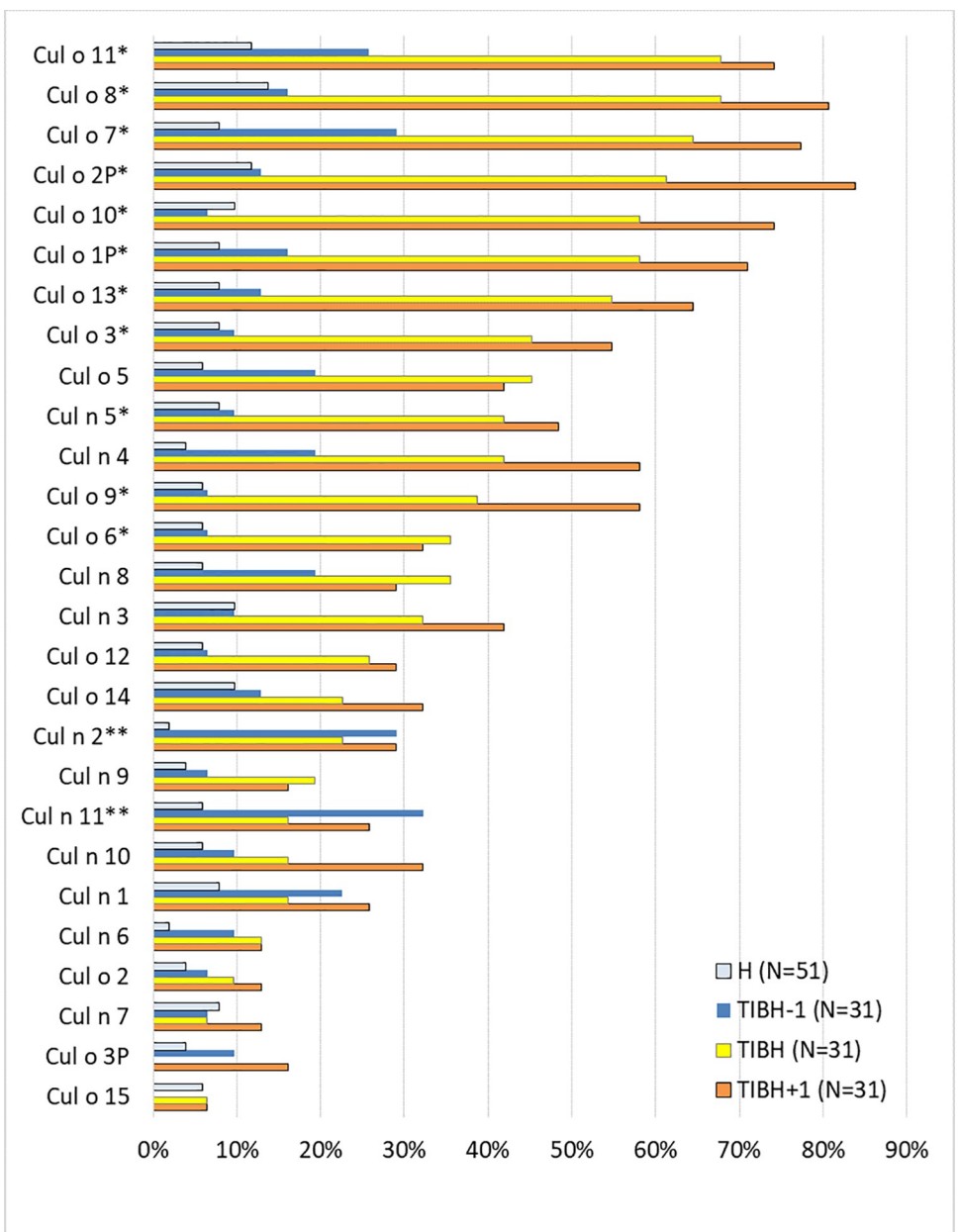

**Fig 3. Percentage of horses with IgE levels above the cut-off values following import from Iceland to Switzerland.** Horses with IgE levels above the cut-off values (as indicated in Table 2) at time of clinical onset of IBH ($T_{IBH}$), one year before ($T_{IBH}-1$)) and one year after ($T_{IBH}+1$) and in healthy horses for comparison. P values were calculated with the Fisher's exact test and Bonferroni correction done for multiple comparisons. * significant difference between $T_{IBH}$ and $T_{IBH-1}$ ($cP < 0.05$). ** significant difference between $T_{IBH-1}$ and H ($cP < 0.05$).

Comparison of IgE concentrations between the groups showed that for most r-allergens IgE levels do not differ significantly depending on duration of IBH. Nevertheless, horses with a long duration of IBH (>7 years) have significantly higher IgE levels to Cul o 1P, Cul o 5 and Cul o 10 compared with horses suffering from IBH for <4 years (Table 5).

Effect of IBH duration on median IgE levels (in fluorescence arbitrary units) to *Culicoides* r-allergens in horses living in Sweden. Same superscript letters indicate statistically significant

**Table 3. Median serum IgE levels to *Culicoides* recombinant (r-) allergens.**

| Allergen name | $T_{IBH}$-1 (n = 31) | | $T_{IBH}$ (n = 31) | | $T_{IBH}$+1 (n = 31) | |
|---|---|---|---|---|---|---|
| | median | range | median | range | median | range |
| Cul o 1P | 1871[a] | 0–50731 | 13269[a,b] | 0–59314 | 45226[b] | 0–55123 |
| Cul o 2 | 600 | 0–179 | 944 | 0–763 | 1338 | 0–263 |
| Cul o 2P | 52[a] | 0–3143 | 471[a] | 0–23064 | 1214 | 0–8783 |
| Cul o 3 | 332[a] | 34–5643 | 905[a] | 82–18971 | 1149 | 210–6384 |
| Cul o 3P | 118 | 0–2031 | 103 | 0–1584 | 189 | 0–21848 |
| Cul o 5 | 2655 | 0–49677 | 4937 | 564–55803 | 5690 | 419–56749 |
| Cul o 6 | 43[a] | 0–1782 | 91[a] | 0–3603 | 134 | 0–2031 |
| Cul o 7 | 353[a] | 64–43021 | 2148[a] | 111–50627 | 4457 | 77–60346 |
| Cul o 8 | 3940[a] | 0–57708 | 46484[a,b] | 0–60605 | 48671[b] | 2479–60389 |
| Cul o 9 | 59[a] | 0–50264 | 485[a,b] | 0–61076 | 3988[b] | 0–63029 |
| Cul o 10 | 32[a] | 0–3311 | 1220[a] | 0–48437 | 1952 | 0–49432 |
| Cul o 11 | 7267[a] | 1457–47315 | 29123[a] | 1035–58052 | 26223 | 1534–59117 |
| Cul o 12 | 11[a] | 0–528 | 76[a] | 0–5598 | 249 | 0–2817 |
| Cul o 13 | 1825[a] | 150–34346 | 6641[a] | 61–48342 | 8935 | 318–53397 |
| Cul o 14 | 0 | 0–347 | 29 | 0–1062 | 36 | 0–2493 |
| Cul o 15 | 66[a] | 0–863 | 135[a] | 0–5573 | 166 | 0–1310 |
| Cul n 1 | 728 | 0–12989 | 590 | 0–40387 | 1007 | 0–45461 |
| Cul n 2 | 108 | 0–530 | 68 | 0–3049 | 90 | 0–2536 |
| Cul n 3 | 672[a] | 100–29127 | 3349[a] | 101–59729 | 4763 | 85–57789 |
| Cul n 4 | 70[a] | 0–33717 | 531[a] | 0–37001 | 1329 | 0–31529 |
| Cul n 5 | 32[a] | 0–218 | 79[a] | 0–4296 | 97 | 0–1515 |
| Cul n 6 | 102 | 0–3805 | 45 | 0–2087 | 54 | 0–45799 |
| Cul n 7 | 478 | 0–5488 | 351 | 0–5495 | 361 | 0–7782 |
| Cul n 8 | 1185 | 165–9676 | 1856 | 108–15927 | 1595 | 21–16744 |
| Cul n 9 | 619 | 176–18617 | 749 | 38–9550 | 798 | 0–27699 |
| Cul n 10 | 944 | 0–6466 | 664[b] | 6–55321 | 1403[b] | 0–35385 |
| Cul n 11 | 498 | 0–10235 | 315 | 0–9431 | 532 | 0–17096 |

Median serum IgE levels (in fluorescence arbitrary units) to *Culicoides* recombinant (r-) allergens in horses imported from Iceland to Switzerland that developed insect bite hypersensitivity (IBH). Median IgE values in sera taken one year before 1st clinical signs of IBH ($T_{IBH}$-1), the year when clinical signs of IBH were first observed ($T_{IBH}$) and the following year ($T_{IBH}$+1) were compared using the non-parametric paired Wilcoxon Signed-Rank Test and Bonferroni corrections for multiple comparisons (cP).

[a] Significant difference between $T_{IBH}$-1 and $T_{IBH}$.

[b] Significant difference between $T_{IBH}$ and $T_{IBH}$+1.

differences (P ≤ .05) in Kruskal-Wallis Z-value test (Dunn's test) with Bonferroni correction for multiple comparisons.

## Discussion

The aim of this study was to identify the primary sensitizing *Culicoides* allergens for equine insect bite hypersensitivity in horses exported from Iceland to continental Europe. Therefore, allergen-specific serum IgE was measured with a newly developed protein microarray [10, 22] which includes a comprehensive panel of 16 Cul o and 11 Cul n r-allergens. As previous studies have shown that allergens derived from *Culicoides* species found in the horses' environment (i.e. *Culicoides obsoletus*) are more relevant for IBH than laboratory-bred species (i.e. *Culicoides nubeculosus*) [14, 25], it was crucial to test a large panel of Cul o r-allergens. The findings

**Table 4. Effect of the origin of the horse on median values of *Culicoides*-specific IgE levels.**

| Allergen | IS-H (N = 13) | | IS-IBH (N = 44) | | N-IS-H (N = 24) | | N-IS-IBH (N = 11) | |
|---|---|---|---|---|---|---|---|---|
| | median | range | median | range | median | range | median | range |
| Cul o 1P | 491 | 0–14889 | 47128 | 169–59261 | 906 | 0–7029 | 42471 | 4026–52188 |
| Cul o 2 | 35 | 0–234 | 168 | 4–6112 | 35 | 0–146 | 36 | 0–2230 |
| Cul o 2P | 53 | 0–199 | 3387 | 37–53231 | 20 | 0–429 | 561 | 108–15757 |
| Cul o 3 | 195 | 32–1067 | 1837 | 125–24957 | 184 | 25–2169 | 526 | 172–5540 |
| Cul o 3P | 446 | 47–3523 | 913 | 48–51070 | 310 | 0–1537 | 610 | 0–2646 |
| Cul o 5 | 523 | 237–22917 | 16522 | 258–53700 | 1048 | 75–8382 | 8104 | 574–47024 |
| Cul o 6 | 91 | 0–417 | 313[a] | 0–12676 | 46 | 0–376 | 66[a] | 0–14442 |
| Cul o 7 | 183 | 18–582 | 15591 | 60–57396 | 358 | 33–2492 | 1795 | 244–15408 |
| Cul o 8 | 3625 | 0–13227 | 46296 | 3647–55048 | 1125 | 46–10171 | 42138 | 1102–50688 |
| Cul o 9 | 76 | 0–1833 | 40984 | 38–56877 | 57 | 0–978 | 10795 | 52–53911 |
| Cul o 10 | 116 | 0–1233 | 20245 | 0–55174 | 204 | 0–612 | 9215 | 107–44976 |
| Cul o 11 | 2608 | 667–34063 | 41605 | 2679–54109 | 1722 | 380–8559 | 28932 | 8193–46491 |
| Cul o 12 | 20 | 0–1017 | 1317 | 0–25360 | 8 | 0–774 | 545 | 0–13677 |
| Cul o 13 | 883 | 74–22480 | 9167[a] | 838–49726 | 700 | 31–3226 | 1121[a] | 123–16088 |
| Cul o 14 | 18 | 0–114 | 190 | 0–2837 | 23 | 0–354 | 38 | 0–2214 |
| Cul o 15 | 76 | 5–513 | 233 | 13–13921 | 68 | 0–1069 | 171 | 0–5898 |
| Cul n 1 | 381 | 1–3269 | 1329 | 184–52050 | 426 | 65–2790 | 831 | 0–48398 |
| Cul n 2 | 96 | 0–430 | 234 | 0–4793 | 50 | 0–363 | 67 | 0–1023 |
| Cul n 3 | 1119 | 211–2192 | 11763 | 344–56358 | 1499 | 134–24146 | 5486 | 862–12438 |
| Cul n 4 | 81 | 0–556 | 2266 | 121–54464 | 152 | 0–871 | 468 | 0–17197 |
| Cul n 5 | 54 | 0–117 | 227[a] | 0–4777 | 9 | 0–108 | 24[a] | 0–179 |
| Cul n 6 | 228 | 0–2142 | 746 | 0–38382 | 180 | 0–1406 | 412 | 0–2303 |
| Cul n 7 | 659 | 0–4642 | 462 | 0–23991 | 292 | 0–4447 | 242 | 0–983 |
| Cul n 8 | 967 | 139–3462 | 1173 | 5–19481 | 1120 | 0–8281 | 1788 | 331–9573 |
| Cul n 9 | 660 | 71–5671 | 1824 | 137–51165 | 457 | 158–13932 | 2159 | 178–21507 |
| Cul n 10 | 658 | 0–11781 | 1374 | 0–49574 | 477 | 0–5097 | 334 | 57–9265 |
| Cul n 11 | 174 | 0–2461 | 877 | 0–42727 | 292 | 0–1658 | 210 | 0–24667 |
| CO-WBE | 549 | 205–2124 | 2321 | 301–36169 | 482 | 31–1860 | 1703 | 485–30119 |
| CN-TE | 744 | 192–2550 | 1553 | 198–42326 | 464 | 55–1981 | 1498 | 476–7854 |

Effect of horse origin (born in Iceland and exported to Sweden (IS) or born in Sweden and living in Sweden (N-IS)) on median values of *Culicoides*-specific IgE levels (in fluoresence arbitrary unit) in IBH-affected and healthy (H) horses. The Kruskal-Wallis Multiple-Comparison Z-Value test (Dunn's test) with Bonferroni correction was used to analyze differences in allergen-specific IgE levels.

Pink: Significant difference between healthy and IBH IS horses.

light blue: Significant difference between healthy and IBH N-IS horses.

[a] Significant difference between IS-IBH and N-IS IBH horses.

No significant differences were observed between IS-H and N-IS-H horses.

of this study indicate a rise in *Culicoides*-specific IgE concomitant with the initial onset of clinical signs of IBH, and that there is not a single, but many primary sensitizing *Culicoides* allergens. This confirms initial investigations, in which only three Cul o allergens and 10 Cul n r-allergens were included and a smaller number of horses were tested [19]. In the present study, seven major primary sensitizing *C. obsoletus* r-allergens were identified, namely Cul o 8, Cul o 11, Cul o 2P, Cul o 7, Cul o 1P, Cul o 13 and Cul o 10. These r-allergens bound IgE in 51 to 78% of the sera at $T_{IBH}$. Two further r-allergens, Cul o 3 and Cul o 9 also appear to be very important at the start of sensitization as they bound IgE in nearly 50% of sera. All these r-

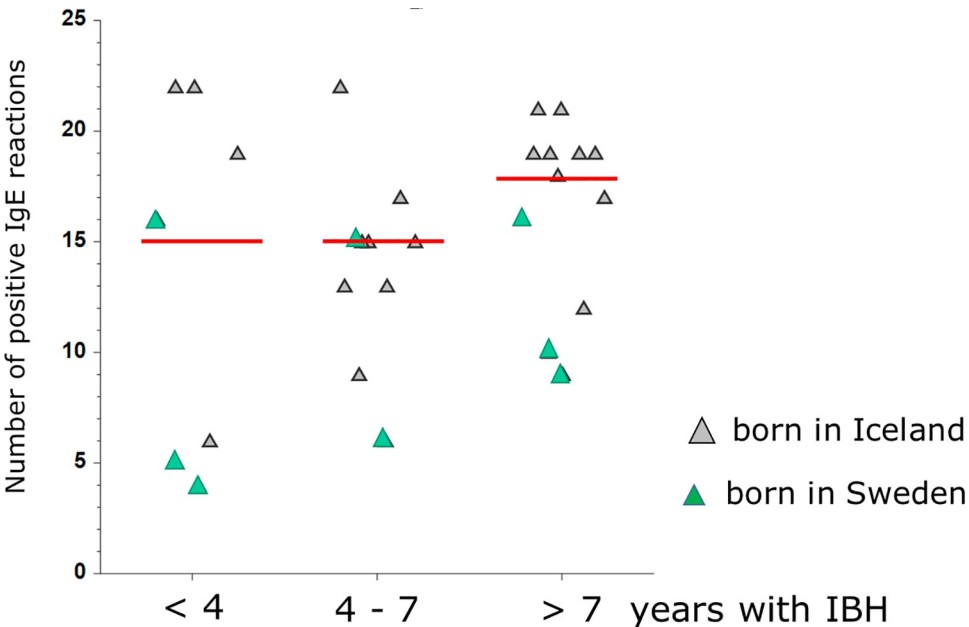

**Fig 4. Effect of duration of IBH on cumulative number of positive serum IgE values.** Icelandic horses living in Sweden grouped according to how long they had been affected with IBH (< 4 years, 4–7 years or >7 years). No significant differences between the groups in the Kruskal-Wallis Multiple-Comparison Z-Value test. Green triangles identify horses that were born in Sweden and grey triangles those that were born in Iceland and exported to Sweden.

allergens, with the exception of Cul o 13, had also previously been identified as those most relevant for IBH in horses from various breeds independent of their origin [10]. Our study indicates that Cul o 13, a D7-related protein, is an important r-allergen in horses exported from Iceland, as >50% of the horses had positive IgE values to this allergen at $T_{IBH}$. Similarly, in the Swedish horse group, IBH horses exported from Iceland had significantly higher IgE levels to this r-allergen than those born in continental Europe. Additionally, a further 10 r-allergens bound IgE more frequently in sera from IBH horses at $T_{IBH}$ compared to the H group, however sero-positivity rates were lower (18–47%, p<0.05).

An overall increase in allergen-specific IgE levels was observed in the second year of IBH symptoms (i.e. between $T_{IBH}$ and $T_{IBH}+1$), although this mostly did not reach statistical significance. Conversely, the number of *Culicoides* r-allergens that horses get sensitized to was not found to increase between $T_{IBH}$ and $T_{IBH}+1$.

The highest increase in serum IgE concentrations occurred between $T_{IBH}-1$ and $T_{IBH}$. At $T_{IBH}-1$ the IBH group did not usually differ significantly from the H group or from the unexposed horses. The sero-positivity rate in the H group was somewhat higher than reported by Novotny et al. [10]. This might be due to the fact that nearly none of the H horses in that study were imported from Iceland. Previous studies showed some degree of sensitization in horses imported from Iceland with a healthy end-point, indicating a mechanism for regulation of this initial sensitization [18]. A small percentage of the unexposed horses had IgE positive values to some of the r-*Culicoides* allergens. This might be explained by endoparasite induced high polyclonal IgE [24, 26] binding nonspecifically, or to cross-reactivity between allergens in *Simulium* and *Culicoides* [11]. *Simulium* are present in Iceland and are the only known insects that bite horses in Iceland.

It will be a great interest to determine in the future whether and which IgG subclasse(s) are associated with a non-allergic immune response to *Culicoides* allergens and have blocking

**Table 5. Effect of duration of IBH on median IgE levels to *Culicoides* r-allergens.**

| Allergen | IBH < 4 years (N = 7) | | IBH 4–7 years (N = 9) | | IBH > 7 years (N = 13) | |
|---|---|---|---|---|---|---|
| | median | range | median | range | median | range |
| Cul o 1P | 27020 [a,b] | 664–51471 | 49206 [a] | 42471–59261 | 48674 [b] | 41046–56746 |
| Cul o 2 | 234 | 0–6112 | 87 | 0–660 | 270 | 0–3949 |
| Cul o 2P | 387 | 108–5277 | 657 | 70–53231 | 2364 | 185–13293 |
| Cul o 3 | 2092 | 344–7757 | 938 | 172–10148 | 1732 | 285–24957 |
| Cul o 3P | 968 | 0–5997 | 559 | 0–2521 | 1096 | 569–20970 |
| Cul o 5 | 5984 [b] | 1748–32944 | 27955 | 1385–47024 | 30804 [b] | 1890–49394 |
| Cul o 6 | 316 | 4–14442 | 515 | 0–3086 | 310 | 9–5993 |
| Cul o 7 | 2289 | 946–55017 | 5658 | 721–41237 | 26305 | 244–53549 |
| Cul o 8 | 42747 | 3647–47801 | 45143 | 19340–55048 | 48555 | 34517–54166 |
| Cul o 9 | 11495 | 52–51561 | 42839 | 7787–55000 | 41388 | 5875–56877 |
| Cul o 10 | 5356 [b] | 107–34307 | 13292 | 4276–44897 | 23654 [b] | 8690–55174 |
| Cul o 11 | 39026 | 15278–47110 | 31165 | 8427–48983 | 41621 | 5414–54109 |
| Cul o 12 | 445 | 0–1978 | 1041 | 137–1863 | 1277 | 190–13677 |
| Cul o 13 | 1451 | 123–45650 | 3198 | 479–49726 | 22463 | 850–48735 |
| Cul o 14 | 197 | 0–2214 | 56 | 0–488 | 134 | 0–2295 |
| Cul o 15 | 96 | 0–4396 | 302 | 0–6080 | 348 | 13–5898 |
| Cul n 1 | 597 | 0–41966 | 1349 | 780–48398 | 4210 | 399–47084 |
| Cul n 2 | 239 | 0–4793 | 192 | 0–673 | 230 | 29–1408 |
| Cul n 3 | 3020 | 844–45049 | 5824 | 344–52551 | 17084 | 657–54196 |
| Cul n 4 | 286 | 45–48935 | 751 | 0–17197 | 2001 | 121–36137 |
| Cul n 5 | 118 | 0–2626 | 47 | 2–2277 | 171 | 1–4777 |
| Cul n 6 | 704 | 0–12758 | 454 | 150–1788 | 1270 | 412–38382 |
| Cul n 7 | 1179 | 0–3069 | 234 | 0–1490 | 421 | 0–7609 |
| Cul n 8 | 2293 | 737–7875 | 2230 | 114–3681 | 1073 | 331–13366 |
| Cul n 9 | 2674 | 305–7540 | 1190 | 423–15522 | 3512 | 402–47688 |
| Cul n 10 | 1252 | 58–12871 | 1153 | 372–49574 | 1835 | 164–45219 |
| Cul n 11 | 138 | 0–28577 | 1184 | 60–24667 | 1539 | 51–42727 |

properties [27]. Previous studies have shown that horses exposed to *Culicoides* bites which do not develop IBH are not immunologically ignorant to these antigens but have an antigen specific Th1/Treg immune response [28–30]. Horses have seven IgG subclasses [31] and, unfortunately, reagents to determine each of the seven IgG subclasses individually are still missing. Previous studies have shown that IgG5 and IgG3/5 to some *Culicoides* r-allergen are increased in serum of horses with IBH, sometimes even before onset of IBH and might thus have a predictive value [19, 32]. However, no protective IgG subclass has been identified yet [19].

Data from the horses living in Switzerland and in Sweden were analyzed separately due to confounding factors. The horses in Sweden had been exposed to *Culicoides* for a longer time period than the Swiss group, so it is unknown whether time of exposure, type or quantity of insects were responsible for the higher degree of IgE sensitization observed. Analysis of the data from the Swedish group indicates that in horses imported from Iceland several years ago that did not develop IBH, IgE levels were similar to those of H horses born in continental Europe. Hence, for horses not susceptible to *Culicoides* allergy, the provenance is not important. On the other hand, for the susceptible ones within the same breed, there is a clear difference in the sensitization level between IS and N-IS IBH horses: IS horses are sensitized to markedly more *Culicoides* r-allergens and often have higher IgE levels (significant for Cul o 6, Cul o 13 and Cul n 5) than IBH horses born in continental Europe. However, this needs to be

confirmed, as the number of IBH horses in the N-IS group was rather small. Nevertheless, this supports our hypothesis that the high degree of sensitization and the high prevalence of IBH in Icelandic horses is not due to the breed itself, but to the presence or absence of *Culicoides* in the environment at early age [4].

Finally, we also investigated whether a longer duration of IBH was associated with sensitization to a higher number of *Culicoides* allergens and/or resulted in higher IgE concentrations to some of the r-allergens. Our data suggests that the number of allergens horses are sensitized to is only slightly higher in horses affected with IBH for many years (>7) than in those affected for a shorter period of time (≤ 7years) (median = 18 versus 15 r-allergens, respectively, ns). Horses with a long history of IBH often had higher *Culicoides*-specific IgE levels, but this difference only reached significance for three r-allergens: Cul o 1P, Cul o 5 and Cul o 10. From our data there is no indication that IgE sensitization decreases over time, even though a reduction of exposure to *Culicoides* through management techniques, such as stabling or use of blankets, is usually done in order to reduce clinical signs of IBH. This is a limitation of this part of the study, as, beside the relatively small groups available to evaluate effects of duration of the disease, treatments could not be accounted for. Furthermore, because of the individual sensitization pattern, a longitudinal study would be more suitable to evaluate such effects.

In conclusion, this study demonstrates that there is no single primary sensitizing *Culicoides* r-allergen, but that horses become sensitized simultaneously to multiple *Culicoides* allergens. This indicates that IgE-reactivity is probably due to co-sensitization as oppose to cross-reactivity between *Culicoides* allergens. The study has enabled the identification of the most relevant primary sensitizing allergens for IBH in horses exported from Iceland to continental Europe. This is a first important step towards the development of preventive allergen immunotherapy for IBH.

## Supporting information

**S1 Table. Median serum IgE levels to *Culicoides* recombinant (r-)allergens.** Median serum IgE levels in fluorescence arbitrary units to *Culicoides* recombinant (r-) allergens in horses imported from Iceland to Switzerland that developed insect bite hypersensitivity (IBH) or remained healthy (H), and in horses living in Iceland (unexposed). Serum samples were taken the summer of clinical onset of IBH ($T_{IBH}$) and at the corresponding time in the H group. Same superscript letters indicate statistically significant differences in Kruskal-Wallis Z-value test (Dunn's test) with Bonferroni correction for multiple comparisons.
(DOCX)

## Acknowledgments

We would like to thank Shui Ling Chu and Jelena Mirkovitch, Department of Clinical Research VPH, Vetsuisse Faculty, University of Bern, for their excellent technical support.

## Author Contributions

**Conceptualization:** Sigurbjörg Torsteinsdottir, Eliane Marti.

**Formal analysis:** Jasmin Birras.

**Funding acquisition:** Sigridur Jonsdottir, Sigurbjörg Torsteinsdottir, Marcos Alcocer, Eliane Marti.

**Investigation:** Jasmin Birras, Samuel J. White, Ella N. Novotny.

**Methodology:** Samuel J. White, A. Douglas Wilson, Marcos Alcocer, Eliane Marti.

**Project administration:** Eliane Marti.

**Resources:** Samuel J. White, Sigridur Jonsdottir, Anja Ziegler, A. Douglas Wilson, Rebecka Frey, Sigurbjörg Torsteinsdottir, Marcos Alcocer, Eliane Marti.

**Supervision:** Eliane Marti.

**Validation:** Jasmin Birras.

**Visualization:** Jasmin Birras.

**Writing – original draft:** Jasmin Birras.

**Writing – review & editing:** Samuel J. White, Sigridur Jonsdottir, Ella N. Novotny, Anja Ziegler, A. Douglas Wilson, Rebecka Frey, Sigurbjörg Torsteinsdottir, Marcos Alcocer, Eliane Marti.

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
