## [Decision Letter · Decision Letter 0]

26 Jul 2021

PONE-D-21-15096

First clinical expression of equine insect bite hypersensitivity is associated with co-sensitization to multiple Culicoides allergens

PLOS ONE

Dear Dr. Marti,

Thank you for submitting your manuscript to PLOS ONE. After careful consideration, we feel that it has merit but does not fully meet PLOS ONE’s publication criteria as it currently stands. Therefore, we invite you to submit a revised version of the manuscript that addresses the points raised during the review process.

Both reviewers are enthusiastic about your manuscript, however, questions were raised that need to be addressed. In particular, reviewer 1 points out that not all responses to allergens would involve IgE.

We look forward to receiving your revised manuscript.

Kind regards,

Ulrike Gertrud Munderloh, Ph.D.

Academic Editor

PLOS ONE

2. In your Methods section, please provide additional details regarding participant consent from the owners of the animals. In the ethics statement in the Methods and online submission information, please ensure that you have specified (1) whether consent was informed and (2) what type you obtained (for instance, written or verbal). If the need for consent was waived by the ethics committee, please include this information.

“This study was supported by the Swiss National Science Foundation grant no 310030-160196/1, by the Morris Animal Foundation grant no D16EQ-039 and by the Stiftung Forschung für das Pferd.”

“This study was supported by the Swiss National Science Foundation grant no 310030-160196/1, by the Morris Animal Foundation grant no D16EQ-039 and by the Stiftung Forschung für das Pferd.”

Additional Editor Comments (if provided):

Reviewers' comments:

Reviewer's Responses to Questions

**Comments to the Author**

1. Is the manuscript technically sound, and do the data support the conclusions?

Reviewer #1: Partly

Reviewer #2: Yes

2. Has the statistical analysis been performed appropriately and rigorously? 

Reviewer #1: Yes

Reviewer #2: Yes

3. Have the authors made all data underlying the findings in their manuscript fully available?

Reviewer #1: Yes

Reviewer #2: Yes

4. Is the manuscript presented in an intelligible fashion and written in standard English?

Reviewer #1: Yes

Reviewer #2: Yes

5. Review Comments to the Author

Reviewer #1: In the manuscript “First clinical expression of equine insect bite hypersensitivity is associated with cosensitization to multiple Culicoides allergens” by Birras et al. the authors explored in a longitudinal study the antigen-specificity of IgE responses to known Culicoides allergens in the serum of horses of islandic breed.

The authors found that almost all horses had shared responses to distinct insect-derived antigens. As responses to most of these antigens were neglectable the season prior to the onset of Insect Bite Hypersensitivity (IBH), the authors conclude that there must be a co-induction to all these antigens at the same time point, during the manifestation of the disease. Alternatively, a dominant antigen could have been the first stimulus that induced an allergic response and then allergic immune responses to all other antigens could have been induced via “epitope spreading”, as it has been shown for many other allergies and auto-immune responses.

Altogether, this is a very nice manuscript, of great interest to the overall scientific community interested in the underlying immunological mechanisms leading to allergies and IBH in specific.

There is just one aspect I would like to see addressed. This is the frequency and strength of allergen-specific non-IgE antibody responses in these animals.

The authors exclusively measured IgE titres in the serum of horses suffering from IBH or not. The half-life of IgE in serum is very short, while IgE remains bound to Ig-ε-R expressing cells for a very long period of time. Thus, serum levels of IgE could easily be misguiding. Furthermore, for many allergens there might also be non-allergic immune reactions, which could induce the expression of allergen-specific IgG antibodies. Such antibodies could potentially outcompete allergen-specific IgE antibodies for binding to the microarray of recombinant antigens, as it has been used in this study. In addition, such antibodies may have a curative effect (e.g. IgG4 antibodies) or may prevent the onset of IBH. For instance, Meulenbroeks et al. (PMID: 25901733) have shown that even horses not affected by IBH are not necessarily immunologically ignorant but have a protective Th1-skewed allergen-specific immune response. Such an alternative explanation could not be ruled out by the presented data. Hence, it appears warranted to complement the current data with total antibody and in particular with allergen-specific IgG antibody titres.

Minor aspect:

Please consistently add “continental” Europe, when referring to Sweden or Switzerland.

Reviewer #2: The paper provides novel information on the development of Equine Insect bite hypersensitivity (IBH) that could be important for the establishment of therapy and prevention of this type of allergy in the future..

The study includes a large number of horses inside every group studied (224 in total) and a control group. A follow-up of three years (enough to obtain good conclusions about the development of the IBH) has been made.

The statistical analyse is well described and is adequate for the data (not normally distribution) and number of animals.

The technique used for the evaluation of the IgE microarray proteins allows the detection of antigenic protein molecules with more precise results than other immunoassay systems used to date and is well described by the authors.

The paper manages to achieve the proposed objectives.

The discussion could be enriched with a broader analysis of the knowledge situation of the IBH so far, but it is understood that the authors have not wanted to enter into comparative aspects because the microarray technique is totally different and much more specific than others used up to now.

I consider that it is suitable to be published.

One error should be corrected on line 40 (same as in the first presentation summary): (<4 years, 4-7 years, > 7 years).

6. PLOS authors have the option to publish the peer review history of their article (what does this mean?). If published, this will include your full peer review and any attached files.

Reviewer #1: No

Reviewer #2: **Yes: **Maite Verde, Veterinary Internal Medicine and Dermatology, University of Zaragoza.

---

## [Author Response · Author response to Decision Letter 0]

16 Aug 2021

Dear Editor, dear Reviewers, 

We would like to thank you for the positive review of our manuscript and for the useful corrections. We have amended our manuscript according to your suggestions and hope that it will now be suitable for publication. 

You can find our response to your comments in blue in the text below and the corrections to the manuscript are marked in yellow. 

On Behalf of the authors

Kind regards,

Eliane Marti

Our response to the reviewers is also provided as a separate file. 

We have checked the requirements and made few corrections. 

2. In your Methods section, please provide additional details regarding participant consent from the owners of the animals. In the ethics statement in the Methods and online submission information, please ensure that you have specified (1) whether consent was informed and (2) what type you obtained (for instance, written or verbal). If the need for consent was waived by the ethics committee, please include this information.

Owner consent information is now provided in lines 149-150.

“This study was supported by the Swiss National Science Foundation grant no 310030-160196/1, by the Morris Animal Foundation grant no D16EQ-039 and by the Stiftung Forschung für das Pferd.”

“This study was supported by the Swiss National Science Foundation grant no 310030-160196/1, by the Morris Animal Foundation grant no D16EQ-039 and by the Stiftung Forschung für das Pferd.”

The funding information has been removed from the manuscript.

Funding statement: 

“This study was supported by the Swiss National Science Foundation grant no 310030-160196/1, by the Morris Animal Foundation grant no D16EQ-039 and by the Stiftung Forschung für das Pferd.” Please add: “and by the Icelandic Research Fund grant no 141071-05.

No references have been removed. However, as we have revised our manuscript according to the suggestions of the reviewers, we have added the following references: 

27. Shamji MH, Durham SR. Mechanisms of allergen immunotherapy for inhaled allergens and predictive biomarkers. The Journal of allergy and clinical immunology. 2017;140(6):1485-98.

28. Meulenbroeks C, van der Lugt JJ, van der Meide NM, Willemse T, Rutten VP, Zaiss DM. Allergen-Specific Cytokine Polarization Protects Shetland Ponies against Culicoides obsoletus-Induced Insect Bite Hypersensitivity. PloS one. 2015;10(4):e0122090.

29. Hamza E, Akdis CA, Wagner B, Steinbach F, Marti E. In vitro induction of functional allergen-specific CD4+ CD25high Treg cells in horses affected with insect bite hypersensitivity. Clinical and experimental allergy : journal of the British Society for Allergy and Clinical Immunology. 2013;43(8):889-901.

30. Hamza E, Wagner B, Jungi TW, Mirkovitch J, Marti E. Reduced incidence of insect-bite hypersensitivity in Icelandic horses is associated with a down-regulation of interleukin-4 by interleukin-10 and transforming growth factor-beta1. Veterinary immunology and immunopathology. 2008;122(1-2):65-75.

31. Lewis MJ, Wagner B, Woof JM. The different effector function capabilities of the seven equine IgG subclasses have implications for vaccine strategies. Molecular immunology. 2008;45(3):818-27.

32. Raza F, Ivanek R, Freer H, Reiche D, Rose H, Torsteinsdóttir S, et al. Cul o 2 specific IgG3/5 antibodies predicted Culicoides hypersensitivity in a group imported Icelandic horses. BMC veterinary research. 2020;16(1):283.

We have also added a reference describing in detail IgE serology using the microarray with serum from horses.

23. White S, Moore-Colyer M, Marti E, Coüetil L, Hannant D, Richard EA, et al. Development of a comprehensive protein microarray for immunoglobulin E profiling in horses with severe asthma. Journal of veterinary internal medicine. 2019;33(5):2327-35

Additional Editor Comments (if provided):

Reviewers' comments:

Reviewer's Responses to Questions

Comments to the Author

1. Is the manuscript technically sound, and do the data support the conclusions?

Reviewer #1: Partly

Reviewer #2: Yes

2. Has the statistical analysis been performed appropriately and rigorously?

 Reviewer #1: Yes

Reviewer #2: Yes

3. Have the authors made all data underlying the findings in their manuscript fully available?

 Reviewer #1: Yes

Reviewer #2: Yes

 4. Is the manuscript presented in an intelligible fashion and written in standard English?

 Reviewer #1: Yes

Reviewer #2: Yes

 5. Review Comments to the Author

 Reviewer #1: In the manuscript “First clinical expression of equine insect bite hypersensitivity is associated with cosensitization to multiple Culicoides allergens” by Birras et al. the authors explored in a longitudinal study the antigen-specificity of IgE responses to known Culicoides allergens in the serum of horses of islandic breed.

The authors found that almost all horses had shared responses to distinct insect-derived antigens. As responses to most of these antigens were neglectable the season prior to the onset of Insect Bite Hypersensitivity (IBH), the authors conclude that there must be a co-induction to all these antigens at the same time point, during the manifestation of the disease. Alternatively, a dominant antigen could have been the first stimulus that induced an allergic response and then allergic immune responses to all other antigens could have been induced via “epitope spreading”, as it has been shown for many other allergies and auto-immune responses.

Altogether, this is a very nice manuscript, of great interest to the overall scientific community interested in the underlying immunological mechanisms leading to allergies and IBH in specific.

There is just one aspect I would like to see addressed. This is the frequency and strength of allergen-specific non-IgE antibody responses in these animals.

The authors exclusively measured IgE titres in the serum of horses suffering from IBH or not. The half-life of IgE in serum is very short, while IgE remains bound to Ig-ε-R expressing cells for a very long period of time. Thus, serum levels of IgE could easily be misguiding. 

I agree that this a problem inherent to measurement of free serum IgE, which may lead to an underestimation of IgE sensitization. However, long-lived IgE plasma cells seem to contribute to a rather continuous supply of IgE antibodies (Luger EO, Wegmann M, Achatz G, Worm M, Renz H, Radbruch A. Allergy for a lifetime? Allergol Int. 2010 Mar;59(1):1-8. doi: 10.2332/allergolint.10-RAI-0175. PMID: 20186004).

Furthermore, for many allergens there might also be non-allergic immune reactions, which could induce the expression of allergen-specific IgG antibodies. Such antibodies could potentially outcompete allergen-specific IgE antibodies for binding to the microarray of recombinant antigens, as it has been used in this study. In addition, such antibodies may have a curative effect (e.g. IgG4 antibodies) or may prevent the onset of IBH. For instance, Meulenbroeks et al. (PMID: 25901733) have shown that even horses not affected by IBH are not necessarily immunologically ignorant but have a protective Th1-skewed allergen-specific immune response. Such an alternative explanation could not be ruled out by the presented data. Hence, it appears warranted to complement the current data with total antibody and in particular with allergen-specific IgG antibody titres.

I fully agree with this comment and we have added a paragraph on this topic in the discussion (lines 408 to 418). Further studies are needed to better characterize the IgG response of horses to allergens and to determine which equine IgG subclass corresponds to human IgG4.

Minor aspect:

Please consistently add “continental” Europe, when referring to Sweden or Switzerland.

Sweden and Switzerland have been replaced with “continental Europe” when suitable. In some cases, it was important to distinguish between the Swedish and Swiss group, as the data from these groups has been analysed separately because of confounding effects. 

Reviewer #2: The paper provides novel information on the development of Equine Insect bite hypersensitivity (IBH) that could be important for the establishment of therapy and prevention of this type of allergy in the future.

The study includes a large number of horses inside every group studied (224 in total) and a control group. A follow-up of three years (enough to obtain good conclusions about the development of the IBH) has been made.

The statistical analyse is well described and is adequate for the data (not normally distribution) and number of animals.

The technique used for the evaluation of the IgE microarray proteins allows the detection of antigenic protein molecules with more precise results than other immunoassay systems used to date and is well described by the authors.

The paper manages to achieve the proposed objectives.

The discussion could be enriched with a broader analysis of the knowledge situation of the IBH so far, but it is understood that the authors have not wanted to enter into comparative aspects because the microarray technique is totally different and much more specific than others used up to now.

I consider that it is suitable to be published.

One error should be corrected on line 40 (same as in the first presentation summary): (<4 years, 4-7 years, > 7 years).

Indeed! Thank you for the correction. It has been changed (line 40). 

 6. PLOS authors have the option to publish the peer review history of their article (what does this mean?). If published, this will include your full peer review and any attached files.

Do you want your identity to be public for this peer review? For information about this choice, including consent withdrawal, please see our Privacy Policy.

 Reviewer #1: No

Reviewer #2: Yes: Maite Verde, Veterinary Internal Medicine and Dermatology, University of Zaragoza.

While revising your submission, please upload your figure files to the Preflight Analysis and Conversion Engine (PACE) digital diagnostic 

This has been done and the figures are suitable for publication in PlosOne.

---

## [Editor Report · Decision Letter 1]

13 Sep 2021

First clinical expression of equine insect bite hypersensitivity is associated with co-sensitization to multiple Culicoides allergens

PONE-D-21-15096R1

Dear Dr. Marti,

We’re pleased to inform you that your manuscript has been judged scientifically suitable for publication and will be formally accepted for publication once it meets all outstanding technical requirements.

Kind regards,

Ulrike Gertrud Munderloh, Ph.D.

Academic Editor

PLOS ONE
---

## [Editor Report · Acceptance letter]

5 Nov 2021

PONE-D-21-15096R1 

First clinical expression of equine insect bite hypersensitivity is associated with co-sensitization to multiple *Culicoides* allergens 

Dear Dr. Marti:

I'm pleased to inform you that your manuscript has been deemed suitable for publication in PLOS ONE. Congratulations! Your manuscript is now with our production department. 

Kind regards, 

on behalf of

Dr. Ulrike Gertrud Munderloh 

Academic Editor

PLOS ONE